# Mapping QTL for Mineral Accumulation and Shoot Dry Biomass in Barley under Different Levels of Zinc Supply

**DOI:** 10.3390/ijms241814333

**Published:** 2023-09-20

**Authors:** Waleed Amjad Khan, Beth Penrose, Sergey Shabala, Xueqing Zhang, Fangbin Cao, Meixue Zhou

**Affiliations:** 1Tasmanian Institute of Agriculture, University of Tasmania, Hobart, TAS 7001, Australia; waleed.khan@utas.edu.au (W.A.K.); beth.penrose@utas.edu.au (B.P.); sergey.shabala@utas.edu.au (S.S.); 2International Research Centre for Environmental Membrane Biology, Foshan University, Foshan 528000, China; 3Department of Agronomy, College of Agriculture and Biotechnology, Zhejiang University, Hangzhou 310058, China; 22116188@zju.edu.cn

**Keywords:** double haploid, elements, *Hordeum vulgare*, quantitative trait loci

## Abstract

Zinc (Zn) deficiency is a common limiting factor in agricultural soils, which leads to significant reduction in both the yield and nutritional quality of agricultural produce. Exploring the quantitative trait loci (QTL) for shoot and grain Zn accumulation would help to develop new barley cultivars with greater Zn accumulation efficiency. In this study, two glasshouse experiments were conducted by growing plants under adequate and low Zn supply. From the preliminary screening of ten barley cultivars, Sahara (0.05 mg/pot) and Yerong (0.06 mg/pot) showed the lowest change in shoot Zn accumulation, while Franklin (0.16 mg/pot) had the highest change due to changes in Zn supply for plant growth. Therefore, the double haploid (DH) population derived from Yerong × Franklin was selected to identify QTL for shoot mineral accumulation and biomass production. A major QTL hotspot was detected on chromosome 2H between 31.91 and 73.12 cM encoding genes for regulating shoot mineral accumulations of Zn, Fe, Ca, K and P, and the biomass. Further investigation demonstrated 16 potential candidate genes for mineral accumulation, in addition to a single candidate gene for shoot biomass in the identified QTL region. This study provides a useful resource for enhancing nutritional quality and yield potential in future barley breeding programs.

## 1. Introduction

Plants, animals and humans require at least 17 mineral elements for adequate nutrition [1]. Plants take up most of these minerals from the soil, while humans and animals primarily obtain these nutrients from plant sources [2]. Inadequate intake of essential elements, such as iron (Fe), zinc (Zn), phosphorus (P), potassium (K), calcium (Ca), magnesium (Mg), manganese (Mn), iodine (I), selenium (Se) and copper (Cu) in human diet is the leading (>65%) cause of childhood deaths worldwide [3]. Of all these, Fe and Zn deficiencies are the most prevalent, affecting over three billion people globally [4]. Human Zn malnutrition is often attributed to the cultivation of staple crops (mostly cereals) on marginal Zn soils [5]. Insufficient intake of Zn in humans leads to overall poor health, anaemia, increased morbidity and high mortality rates [6]. In contrast, cadmium (Cd) is a common contaminant in agricultural soils [7,8], which is toxic to both animals and plants [9]. As Cd shares uptake pathways with Zn, and its concentration increases with increased Zn concentration in grains of Zn biofortified wheat cultivars [10], it is crucial to confirm if high Zn accumulating varieties contain unhealthy concentrations of Cd.

Barley (*Hordeum vulgare* L.) belongs to the *Poaceae* (grass) family, and is an important cereal crop worldwide [11]. A large variation in Zn accumulation has been previously reported among barley genotypes [12,13,14]. Zinc accumulation in plants is thought to be regulated by a plethora of mechanisms involving root Zn uptake [15,16], root-to-shoot Zn transport [17], subcellular compartmentalisation of Zn [18], and remobilisation of stored Zn from leaf to developing grain [19]. Several studies have attempted to identify quantitative trait loci (QTL) for Zn accumulation in barley grains; using a biparental mapping population (Clipper and Sahara 3771), some overlapping QTL for grain Zn were identified on chromosomes 2H and 4H while QTL on chromosomes 1H, 3H and 5H were inconsistent [20,21,22]. Another mapping approach using a genome-wide association study (GWAS) has validated the involvement of genomic regions on the chromosomes 2H and 4H in grain Zn regulation [23,24,25]. Several additional QTL associated with Zn accumulation were also detected on chromosomes 1H, 3H, 5H, 6H and 7H [23,24,26,27,28]. Metal tolerance protein (*MTP5*) and squamosa promoter binding-like protein (*SPL*) are the potential candidate genes for increasing Zn accumulation in barley [24,26]. Tiong et al. [29] also demonstrated six genes from the ZIP gene family that are associated with enhanced uptake and root-to-shoot translocation of Zn in barley.

Comprehensive knowledge regarding macro- and micronutrient concentration is important to understand their relationships and assess elemental profile of food crops. However, only a few studies [12,28,30] have focused on the elemental concentrations in barley, and no studies used both adequate and deficient Zn growing conditions to investigate QTL for Zn accumulation. This study used shoot Zn accumulation as a selection criterion for grain Zn to screen barley genotypes, as shoot Zn has been shown to have a significant correlation with grain Zn [14]. This study aimed to: (i) investigate the differences in the accumulation of minerals (including Zn) and shoot biomass among different genotypes under different Zn supply; (ii) identify QTL for the accumulation of minerals (including Zn) and shoot biomass under different Zn supply; and (iii) determine relationships among mineral accumulations and biomass.

## 2. Results

### 2.1. Zinc Accumulation in Barley Cultivars

Initially, the screening of ten barley cultivars was performed to determine the most suitable parental genotypes for QTL identification. Shoot Zn accumulation varied between the cultivars under both adequate and low Zn conditions, ranging from 0.42 to 0.48 mg/pot at adequate conditions (Figure 1). Brindabella was the cultivar with the greatest shoot Zn accumulation at adequate conditions, followed by cv. Zhepi-2 and Clipper, whereas cv. CM72 had the lowest shoot Zn accumulation. The cultivars showed variation in Zn accumulation when grown at different Zn supplies. Sahara (0.05 mg/pot) showed the lowest difference in shoot Zn accumulation followed by cv. Yerong (0.06 mg/pot), whereas the highest difference was observed in cv. Franklin (0.16 mg/pot). This indicates that cv. Sahara and Yerong are probably more tolerant to Zn deficiency than cv. Franklin, which showed a greater change in plant Zn accumulation due to Zn supply.

### 2.2. Shoot Mineral Accumulation and Biomass of Barley Double Haploid Lines

Mineral accumulation of all the studied elements in shoots and biomass production of DH lines, including the parents, are summarised in Table 1 and presented in detail in Appendix A. Normal frequency distribution of mineral accumulation in shoots and biomass was observed. As expected, shoot Zn accumulation was greater under adequate Zn conditions than at low Zn conditions (Figure 2).

Consistent with the results of the preliminary experiment, cv. Yerong showed a higher shoot Zn accumulation than cv. Franklin under −Zn conditions. Under low Zn supply, shoot Zn accumulation in cv. Franklin measured almost half that of the plants grown at adequate Zn conditions. Meanwhile, shoot Zn accumulation in cv. Yerong plants when grown at low Zn supply were only 18.3% less than those at adequate Zn conditions. Likewise, Franklin cultivar showed relatively lower shoot biomass under low Zn (−6.6%) than under adequate Zn conditions, whereas cv. Yerong did not exhibit any biomass reduction.

For the DH population, the average shoot Zn accumulation was found to be 38.9% less under low Zn supply than the plants with adequate Zn. Similarly, the plants supplied with low Zn also had 5.6% less biomass production than plants supplied with adequate Zn (Figure 3).

### 2.3. Correlation between Traits

The correlations between biomass and accumulation of different elements were more consistent across different Zn supplies. Importantly, shoot biomass showed a significant positive relationship with all the element accumulations in shoots (Figure 4). Accumulations of P, K and Ca showed strong positive correlations with each other under both adequate and low Zn supplies (Figure 4). Shoot Mn accumulation had the highest positive correlation (R^2^ = 0.75) with Zn accumulation, suggesting that the mechanisms for uptake and transporters of these minerals might be common in barley [31,32]. Overall, Cd had the lowest correlation with other studied elements.

### 2.4. Identification of QTL

One major QTL each was identified for Zn, Fe, Ca, K and P accumulation and shoot biomass on chromosome 2H between 31.91 and 73.12 cM of barley genome (Table 2; Figure 5); however, this study did not find any QTL for Mn and Cd accumulation. The QTL for shoot biomass and mineral accumulations including Zn, Ca, and P were detected at both +Zn and −Zn conditions, while QTL for Fe and K were only detected at −Zn conditions. The total phenotypic variance explained (PVE) by the QTL ranged from 9.9 to 12.1 for shoot Zn accumulation and 19.3–27.3 for biomass production across the two different Zn levels. The PVEs by the QTL were estimated between 8.9 and 21.1 for the rest of the mineral accumulations in shoots. All the QTL for Zn, Fe, Ca, K and P accumulation were mapped on a hotspot of chromosome 2H between flanking markers of bPb-8750 (31.91 cM) and Bmac0093 (73.12 cM) (Figure 5). A major QTL for shoot biomass was also detected within the same locus on chromosome 2H between flanking markers of bPb-4875 (44.04 cM) and Bmac0093 (73.12 cM). This result suggests that a similar mechanism might be involved in the shoot accumulations of Zn, Fe, Ca, K and P and biomass in barley. A QTL for shoot biomass was also detected on the chromosomal region of 2H. Overall, the QTL for accumulation of the minerals and biomass production showed a greater LOD value at low Zn than adequate Zn conditions, which suggests that the candidate genes regulating mineral accumulation and biomass production are more pronounced under low Zn conditions than adequate Zn.

## 3. Discussion

This study demonstrated strong effects of genotype and Zn supply on the mineral accumulation (especially focusing on Zn) and shoot biomass in barley. Interestingly, the differences in Zn accumulation and biomass yield between cv. Franklin (low Zn accumulating) and Yerong (high Zn accumulating) were detected only at low Zn conditions (Figure 2). This suggests that the molecular mechanisms underlying Zn accumulation and homeostasis may vary between the genotypes [34]. Although cv. Franklin and Yerong did not differ substantially in shoot mineral accumulation and biomass production under adequate Zn conditions, large segregations of these traits were observed in their progenies (Table 2). The frequency distributions of Zn accumulation and biomass yield showed transgression in both directions, suggesting that both parents carried genes with alleles contributing to an increased or decreased accumulation of the traits studied. A previous study also found a similar distribution trend of shoot Zn accumulation in the DH lines of Sahara × Clipper [22], indicating the complexity of molecular mechanism underlying shoot Zn accumulation mechanism, as well as the involvement of strong genotype × environment interaction. These results suggest that, despite the effectiveness of genotypes, the availability of Zn in growth medium remains a limiting factor that may reduce Zn accumulation (up to 38.9%) and shoot biomass (up to 5.6%) in barley (Figure 3).

One major QTL was identified for Zn accumulation on chromosome 2H between 31.91 and 73.12 cM. By comparing the closely linked markers [35], the QTL identified in this study is in a different position to that previously reported which is located at 10–30 cM on 2H [21,22]. The QTL Zn accumulation was also co-located with the QTL for elemental accumulation of Fe, Ca, K and P. This may not be surprising, as maintaining ion homeostasis requires a network of ion uptake, transportation, trafficking, and sequestration mechanisms, and not all genes in this regulatory network will be ion-specific. This present study found substantial colocalization of P QTL with cation QTL. P is a component of key molecules of plants such as ATP, nucleic acids, and the form of P most readily accessed by plants, inorganic P, is likely co-transported with positively charged ions [36]. Colocalization of a P QTL with a cation QTL in our study might thus reflect co-transport of P and cations at the transcriptional level. Few cation transporters annotated for *A. thaliana* were demonstrated previously in the P QTL intervals, including high-affinity K^+^ transporter, ZIP metal ion transporter family, and Ctr copper transporter family [37]. Additionally, this study also identified co-location between QTL for mineral accumulations and biomass. This type of colocalization has been previously demonstrated in rice grains [38]. Alternatively, there is also a possibility that all these colocalizations of QTL are coincidental and/or simply due to multiple linked genes. Therefore, we suggest that further investigations may help for an in-depth understanding of the relationships between genetic factors for elemental and yield-related traits in barley.

The release of plant genomes has enabled the identification of putative ionomic genes [39,40], which has opened the way for their detailed characterisation, with some of the genes already validated for their roles in ion cellular import/export and intracellular trafficking [41,42]. Multiple genes have been previously reported for regulating ion homeostasis in plants, which belong to the Zn and Fe-Regulated Transporter-Like Protein (ZIP), Metal Tolerance Protein (MTP), Heavy Metal ATPase (HMA), Natural Resistance-Associated Macrophage Protein (NRAMP), Vacuolar Iron Transporter (VIT), Yellow Stripe-Like Protein (YSL), basic-loop-helix (bHLH) and ATP-binding cassette (ABC) transporter gene families [34,43,44].

The present study detected 16 potential candidate genes within the identified QTL regions influencing shoot elemental concentrations. These genes include four ZIP, three BHLH, three ABC, three potassium transporters, two HMA and one cation calcium exchanger gene (Appendix A). Previously, Reuscher et al. [45] and Detterbeck, Nagel, Rensch, Weber, Borner, Persson, Schjoerring, Christov and Clemens [24] revealed zinc transporter gene 8 (HORVU2Hr1G025400), a homolog of the *A. thaliana* ZIP1 gene and rice OsZIP3 gene, in the specified region of chromosome 2 at 725,227,365 bp. This gene is a member of ZIP family, which is known to enhance Zn uptake and translocation of Zn between root and shoot. Additionally, two YSL and one MTP gene have been also previously identified in barley, contributing to grain Zn accumulation [24]. Furthermore, IDI7, a Fe-regulated ABC transporter, has been shown to be involved in Fe sequestration in barley roots and leaves [46], and a gene annotated as a member of the BHLH transcription factor family plays a critical role in Zn and Fe homeostasis in barley [26,47]. The HMA gene family is also known to play significant roles in metal trafficking in plants. For example, functional validation of HMA genes revealed their involvement in transporting Zn and Cd into the roots and shoots [48] and the mobilisation of Cu and Zn within the plastids and aleurone cells in barley [49]. Lastly, potassium transporter genes have been previously identified as participating in the uptake and translocation of K^+^ to leaves [50]. In the present study, the potential candidate gene for regulating shoot biomass is flowering-promoting factor 1 (FPF1) [51]. We suggest that the candidate genes identified in this study can be targeted for future fine-mapping research in barley.

The Cd accumulation identified in the plant samples of this study did not exceed the levels (100 µg/kg) considered toxic for human consumption [52,53]. More importantly, only a weak correlation (R^2^ = 0.19) was identified between shoot Zn and Cd accumulation in this study (Figure 4). These results suggest that selecting cultivars based on higher Zn is not likely to increase Cd accumulation in plants, which contradicts the findings of Hussain, Khan and Rengel [10] for wheat grains. There is also a possibility that this relationship might be dependent on other variables, e.g., genotype [54] and soil conditions, such as the pH [55] and organic matter content [56]. This study identified a strong positive correlation (R^2^ = 0.75) between Zn and Mn, which is in agreement with a previous study where strong correlations were also found between those elements in introgression lines of barley [45]. Strong correlations between accumulations of P, Mn, Ca and Fe were also observed. This may be related to the stored forms of P after binding with cationic elements as phytates [57]. Kumar et al. [58] demonstrated that the flag leaves store P and distribute it to the grains through vascular bundles [58]. Moreover, shoot biomass was found to have a significant positive relationship with the accumulation of all the studied elements in shoots. This observation suggests that the pathways underlying K, P, Ca, Zn, Mn and Fe accumulations all contribute to increased shoot biomass.

## 4. Materials and Methods

### 4.1. Plant Material

#### 4.1.1. Experiment 1

Ten barley varieties, Sahara, Clipper, Franklin, Brindabella, CM72, Yerong, TX9425, Skiff, Unicorn, and Zhepi-2, were used to determine their Zn accumulation in shoots. Among them, Sahara and Clipper were used as controls for Zn use efficient and inefficient cultivars, respectively, according to the findings of previous studies [22,59].

#### 4.1.2. Experiment 2

As Franklin and Yerong showed contrasting shoot Zn concentration and biomass in Experiment 1 (Figure 1), a double haploid (DH) population consisting of 176 lines derived from a cross between these two varieties, which are constructed by the anther culture method [60], was used to map the QTL for the accumulations of Zn and other minerals under different Zn supply [35]. All barley seeds used were obtained from the University of Tasmania double haploid collection.

### 4.2. Substrate Material and Experimental Design

Four seeds of each genotype were sown in each pot filled with approximately 320 g sand and perlite mix (70:30) under glasshouse conditions, with mean day/night temperatures of 25/15 °C and relative humidity of 65–75%. Three seedlings were kept after germination. All the pots were placed on a large tray containing nutrient solution to grow plants in semi-hydroponics conditions. Both experiments used a randomised complete block design, where there were three large trays for each of the two Zn treatments: adequate Zn (full strength Hoagland’s solution containing Zn; pH = 7) and low Zn (full-strength Hoagland’s solution minus Zn; pH = 7). Each tray contained a set of all the genotypes, three pots as biological replicates for each genotype × Zn treatment. To minimise bias, placement of the pots for same genotypes was different in each tray. The measured Zn concentrations in the nutrient solutions were 0.5 ppm for low Zn (−Zn) and 5 ppm for adequate Zn conditions (+Zn). The nutrient solution was renewed weekly and aerated with a pump. There were three individual plants per pot and the number of pots for each genotype per treatment (i.e., biological replicate) was three. The design and conditions used in both the experiments of this study were the same. The above-ground parts (whole shoots) of plants were harvested after six weeks (at stage 47 based on Zadok’s scale) and immediately oven dried at 60 °C for 72 h. After drying, the biomass of samples was recorded, then the shoots were ground using a coffee grinder and stored in plastic bags until analysis. Mineral accumulation (Equation (1)) in barley shoots of plants was calculated as follows:(1)Mineral accumulation=shoot dry weight × shoot Zn concentration

### 4.3. Sample Preparation for Elemental Analysis

Approximately 0.1 g of each shoot sample was digested by adding 2 mL of 50% (*v*/*v*) nitric acid (HNO_3_; Fisher Scientific, Hampton, NH, USA), and the samples were then heated at 120 °C for 3 h using an aluminium block heater (Dry ThermoUnit, Taitec Corp., Tokyo, Japan) followed by cooling down to room temperature for 30 min. The digested samples were then diluted to a final volume of 50 mL by adding Milli-Q water. Each sample had three analytical replicates. Finally, P, K, Ca, Mn, Fe, Zn and Cd concentrations were measured using inductively coupled plasma optical emission spectroscopy (ICP-OES; Optima 8000, Perkin Elmer). A certified reference material (CRM; NIST 1547 peach leaves) was included in each digestion batch for quality assurance and the recovery for each of the minerals of interest was above 90% in the CRM.

### 4.4. QTL Mapping and Candidate Gene Search

The biomass and mineral accumulation of P, K, Ca, Mn, Fe, Zn and Cd in shoots were used as traits/indicators to detect the associated QTL in barley. The genetic linkage map assembled by Li and Zhou [61] was used in this study for identifying genetic regions closely associated with traits. This linkage map consists of 28 microsatellites and 196 diversity arrays technology (DArT) markers. QTL mapping was performed using the MapQTL 6.0 software [62], where interval mapping (IM) function was used and a logarithm of the odds (LOD) threshold value of 3 to declare a significant QTL. The QTL were named following the nomenclature recommended by McCouch and Cgsnl [33]. Graphical representations of linkage groups and QTL were carried out using MapChart 2.2 [63]. In addition, the annotation file for barley (Morex V3) was accessed from the Ensembl Plants database: http://plants.ensembl.org/ (last accessed on 13 September 2023). Based on the classification of gene functions, we searched for genes related to mineral accumulation and biomass production located within the 1.5-LOD confidence intervals of the detected QTL that were regarded as candidate genes, as described previously [64].

### 4.5. Statistical Analysis

The frequency distribution of the data was plotted as histograms and the normality was estimated using a Shapiro–Wilk in R [65]. Spearman correlation coefficients between the traits were calculated and constructed using the PerformanceAnalytics R package [66]. The confidence interval was set at *p* < 0.05.

## 5. Conclusions

This study identified a major QTL hotspot for Zn, Fe, Ca and K accumulation and biomass yield in barley shoots. This QTL was stable under both low and adequate Zn conditions, but more pronounced (i.e., greater LOD value) at low Zn supply. There was a considerable decrease in shoot Zn accumulation and biomass when the amount of Zn decreased from adequate to low in the nutrient solution. A positive correlation was demonstrated between all traits studied here in barley shoots, including P, K, Ca, Mn, Fe and Zn accumulation and biomass. The results of this study increase our understanding of the genetic basis for regulating shoot mineral accumulation and biomass in barley.

## Figures and Tables

**Figure 1 ijms-24-14333-f001:**
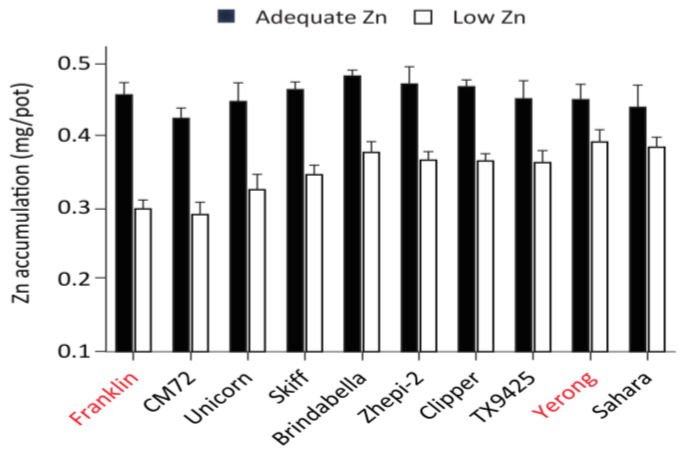
Zinc (Zn) accumulation in shoots of barley cultivars at adequate (black bars) and low Zn (white bars) supply. A mapping population of Yerong × Franklin (shown in red font) was further used for QTL identification. Mean ± SE (n = 3).

**Figure 2 ijms-24-14333-f002:**
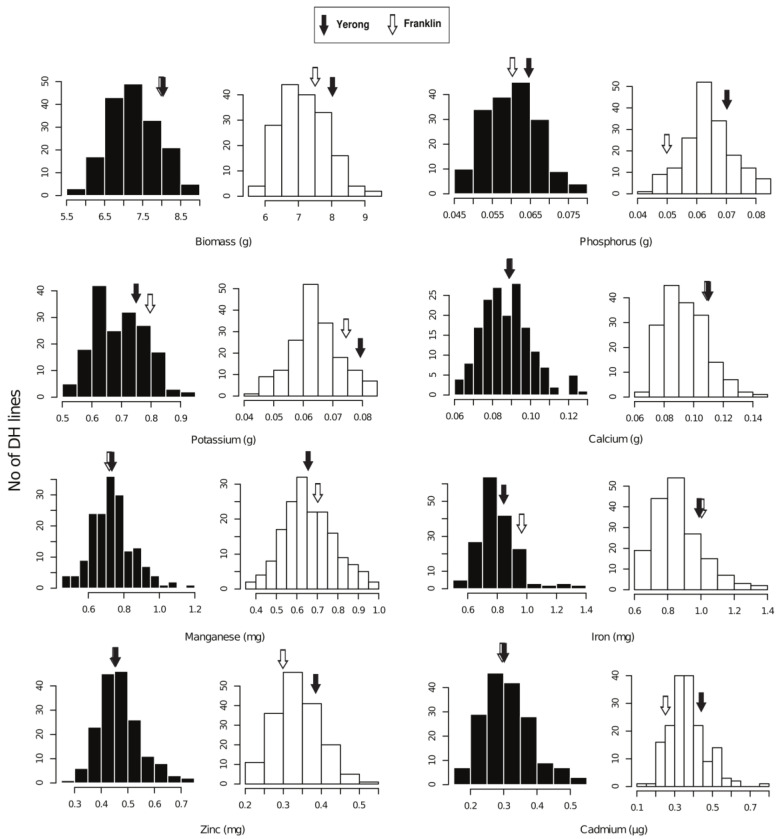
Frequency distributions of the shoot mineral accumulation and dry biomass of the Franklin × Yerong DH population. Plants were supplied with adequate (black bars) and low (white bars) Zn supply in nutrient solution. Accumulation of minerals in shoots and shoot dry biomass is presented per pot containing three individual plants. Arrows indicate the position of both parents for each trait in the histogram.

**Figure 3 ijms-24-14333-f003:**
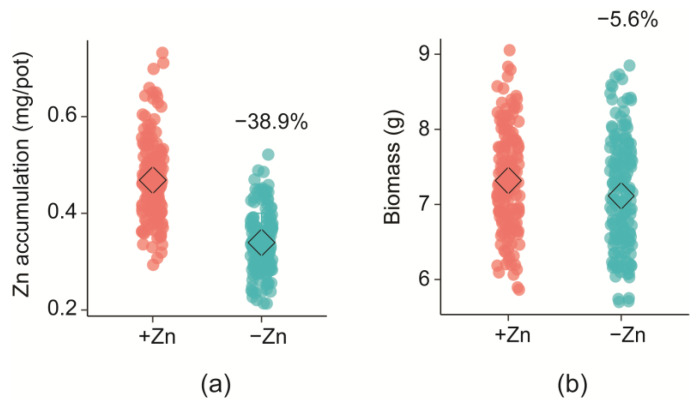
Effect of zinc nutrient supply on (**a**) Zn accumulation and (**b**) biomass production in barley DH lines (Franklin × Yerong). The diamonds in the chart represent the mean of shoot Zn accumulation (**left**) and biomass production (**right**), whereas the number shows the percentage change in shoot Zn accumulation of plants due to the change in the level of Zn supply.

**Figure 4 ijms-24-14333-f004:**
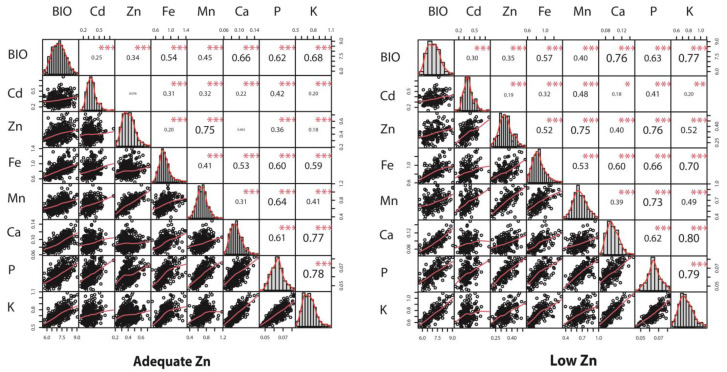
Correlations among mineral accumulations in shoots and dry shoot biomass identified in Franklin × Yerong DH population at adequate and low Zn conditions. BIO: biomass production; (**left**): plants grown under adequate Zn supply; (**right**): plants grown under low Zn supply. * Significant at *p* < 0.05, ** significant at *p* < 0.01, *** significant at *p* < 0.001.

**Figure 5 ijms-24-14333-f005:**
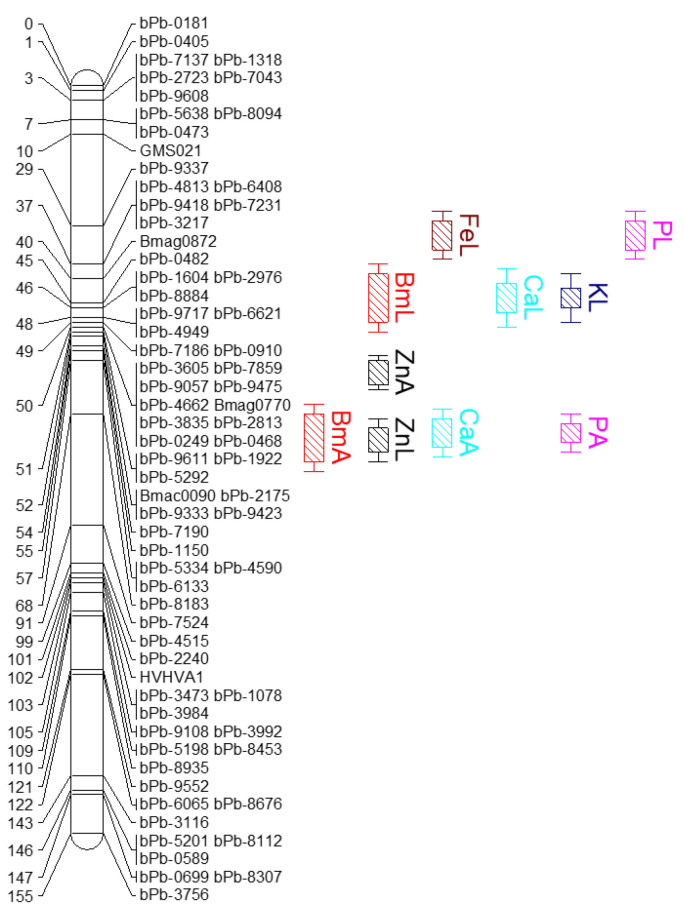
Genetic map of the Franklin × Yerong DH population with identified QTL for shoot biomass and mineral accumulations grown under adequate and low Zn conditions. Bars and letters represent the QTL for the specific trait identified on chromosome 2H of barley. The QTL was named following the nomenclature recommended by McCouch and Cgsnl [33], with initial letters representing the symbol of element and Bm for shoot biomass, whereas the last letters A and L represent adequate and low Zn conditions, respectively, at which QTL is reported.

**Table 1 ijms-24-14333-t001:** Shoot mineral accumulation and dry biomass (per pot) of the parental accessions (Franklin × Yerong) and the DH population in low and adequate Zn conditions. There were three individual plants per pot, and the number of pots for each genotype per treatment (i.e., biological replicate) was 3.

Trait	Zinc Level in Growing Media	DH Parents (Mean)		DH Lines	
		Franklin	Yerong	Range	Mean
Biomass (g/pot)	Adequate	8.00	8.02	5.80–8.99	7.25
	Low	7.50	8.05	5.70–9.04	7.14
Phosphorus (g/pot)	Adequate	0.06	0.07	0.05–0.08	0.06
	Low	0.05	0.07	0.05–0.08	0.06
Potassium (g/pot)	Adequate	0.79	0.76	0.52–0.94	0.70
	Low	0.74	0.78	0.53–1.08	0.74
Calcium (g/pot)	Adequate	0.09	0.09	0.06–0.13	0.09
	Low	0.11	0.11	0.07–0.15	0.09
Manganese (mg/pot)	Adequate	0.75	0.77	0.56–1.33	0.79
	Low	0.71	0.67	0.37–0.93	0.65
Iron (mg/pot)	Adequate	0.88	0.84	0.56–1.33	0.79
	Low	1.05	0.91	0.61–1.31	0.84
Zinc (mg/pot)	Adequate	0.46	0.46	0.29–0.73	0.47
	Low	0.30	0.38	0.21–0.52	0.34
Cadmium (μg/pot)	Adequate	0.30	0.30	0.17–0.55	0.31
	Low	0.26	0.46	0.14–0.77	0.37

**Table 2 ijms-24-14333-t002:** QTL for mineral accumulation and shoot biomass detected in the Franklin × Yerong DH population under low and adequate Zn conditions.

Trait	Chr	Position (cM)	Closest Marker	LOD	R^2^ (%)
BmA	2H	73.12	Bmac0093	7.92	19.3
BmL	2H	44.04	bPb-4875	11.77	27.3
ZnA	2H	59.15	bPb-3572	3.79	9.9
ZnL	2H	72.67	Bmag0518	4.63	12.1
FeL	2H	31.91	bPb-8750	3.45	8.9
CaA	2H	73.12	Bmac0093	4.52	11.5
CaL	2H	44.24	bPb-9682	8.76	21.1
KL	2H	44.24	bPb-9682	7.75	18.9
PA	2H	71.58	bPb-3056	5.45	13.7
PL	2H	31.97	bPb-8750	4.48	11.4

The letter (L or A) in QTL name indicates substrate Zn concentrations: A = adequate, L = low, and Bm is shoot biomass.

## Data Availability

The data presented in this study are available upon request from the corresponding author.

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
