# Peer review of "Mapping QTL for Mineral Accumulation and Shoot Dry Biomass in Barley under Different Levels of Zinc Supply"

_ijms, 2023, doi:10.3390/ijms241814333_

Round 1

Reviewer 1 Report

The authors investigated differences in biomass productivity and accumulation of Zn and other minerals in shoot tissues from barley genotypes. They conducted the experiment under two conditions with different levels of Zn, which is a uniqueness of their study. Additionally, they searched QTLs underlying the differences between two cultivars (Franklin and Yerong) under the two conditions using their progenies. They cloned QTLs which associate with biomass productivity and Zn, Fe, Ca, K, and P accumulation on the chromosome 2H. The QTLs did not associate with Cd accumulation. Gene mapping will be performed in future. 

In the introduction part, it is stated that the goal of this study is to increase the Zn nutritional quality of food crops including barley. The authors analyzed shoot samples to screen barley genotypes and clone QTLs in the present study because it has been reported that shoot Zn accumulation positively correlates with grain Zn. However, they did not further analyze the effects of QTLs on Zn in barley grains. The analysis of grains is almost essential in this study.

In addition, as also stated in the introduction part, QTLs and responsible genes for Zn accumulation in barley grains have been identified in earlier studies. Are these QTLs and genes overlapped with the QTLs cloned in the present study? Is there difference in the effects of QTLs?? What is a novelty and/or uniqueness regarding the QTLs cloned in the present study??? The authors need to discuss about these points more in details.

Furthermore, I would ask the authors to consider the following points.

Abstract

P1L13-14: “quantitative trait loci (QTL)” → quantitative trait loci (QTLs)

Introduction

P1L29: For plants, there are 14 soil essential nutrients (elements), but additional 3 elements, i.e., C, H, and O are also essential. For animals, the number of essential elements should be different.

P1L37: “Inadequate consumption”  Insufficient intake (?)

P2L50: What “consistent” means? Were the same (overlapped) QTLs identified on the chromosomes 2H and 4H??

Results

P5L127: It is difficult to suggest co-transportation of Mn and Zn only by their positive correlation in accumulation. Please cite a literature if the authors mention about the co-transportation.

P5L135: Were QTLs detected only on the chromosome 2H? Or was the QTL analysis performed only on the chromosome 2H (if it so, why)??

Discussion

P7L170: What is “39”?

P7L175-178: “A previous study …”  Please cite the paper.

P8L208: How the candidate genes were determined? The authors should describe about it in the materials and methods part.

Materials and Methods

P9L255: Please describe the method to produce the DH lines (or cite a literature describing the method in details).

Author Response

Dear Reviewer,

Thanks for your constructive suggestions. We have now revised the MS according to your suggestions. All revised parts are in red font. Below are responses to your comments.

Reviewer #1:

In the introduction part, it is stated that the goal of this study is to increase the Zn nutritional quality of food crops including barley. The authors analysed shoot samples to screen barley genotypes and clone QTLs in the present study because it has been reported that shoot Zn accumulation positively correlates with grain Zn. However, they did not further analyze the effects of QTLs on Zn in barley grains. The analysis of grains is almost essential in this study.

Explained: We previously published a paper where we conducted a meta-analysis based on published literature on barley (Khan, W.A.; Shabala, S.; Cuin, T.A.; Zhou, M.; Penrose, B. Avenues for biofortification of zinc in barley for human and animal health: a meta-analysis. Plant Soil 2021, 466, 101–119, doi:10.1007/s11104-021-05027-3). In that paper, we demonstrated a significant positive correlation between grain zinc and shoot zinc, which showed that shoot zinc accumulation could be used as a proxy for grain zinc. Therefore, in this study, we only focused on measuring elemental accumulations in shoots of barley (Line 64 - 65 in the text).

In addition, as also stated in the introduction part, QTLs and responsible genes for Zn accumulation in barley grains have been identified in earlier studies. Are these QTLs and genes overlapped with the QTLs cloned in the present study? Is there difference in the effects of QTLs?? What is a novelty and/or uniqueness regarding the QTLs cloned in the present study??? The authors need to discuss about these points more in details.

Reply: Thanks for the suggestion (we should have done the comparison). Details have been added (Line 200 – 203).

Abstract:

P1L13-14: “quantitative trait loci (QTL)” → quantitative trait loci (QTLs)

Explained: Both QTL and QTLs are acceptable and we prefer to use QTL for both quantitative trait loci and quantitative trait locus.

Introduction:

P1L29: For plants, there are 14 soil essential nutrients (elements), but additional 3 elements, i.e., C, H, and O are also essential. For animals, the number of essential elements should be different.

Reply: We rephrased this sentence and the number of essential elements has been changed to 17 from 14 (Line 28).

P1L37: “Inadequate consumption” → Insufficient intake (?)

Reply: This word has been replaced accordingly in the revised version (Line 36)

P2L50: What “consistent” means? Were the same (overlapped) QTLs identified on the chromosomes 2H and 4H??

Reply: Yes, they were overlapping QTLs identified on the chromosomes 2H and 4H. We have replaced the word ‘consistent’ with ‘overlapping’ for a better understanding of the meaning there (Line 49)

Results

P5L127: It is difficult to suggest co-transportation of Mn and Zn only by their positive correlation in accumulation. Please cite a literature if the authors mention about the co-transportation.

Reply: We have cited two references in the text (Line 136).

Discussion

P7L170: What is “39”?

Reply: Corrected, should be the reference [34].

P7L175-178: “A previous study …” → Please cite the paper.

Reply: This reference has been added to the text of the revised version (line 193)

P8L208: How the candidate genes were determined? The authors should describe about it in the materials and methods part.

Reply: The annotation file for barley (Morex V3) was accessed from the Ensembl Plants database: https:// http://plants.ensembl.org/ (last accessed in Dec, 2022). Based on the classification of gene functions, we searched for genes related to mineral accumulation and biomass production located within the 1.5-LOD confidence intervals of the detected QTL were regarded as candidate genes as described previously [61]. This information has been added to the text of the revised manuscript (line 324 - 327).

Materials and Methods

P9L255: Please describe the method to produce the DH lines (or cite a literature describing the method in details).

Reply: The method and a reference are added (line 279-280).

Reviewer 2 Report

Dear Editors,

Thank you for choosing me as a reviewer of the of the manuscript: Plant Stress-Induced Responses and Tolerance Mechanisms at Biochemical, Cellular, Physiological and Molecular Levels” submitted to IJMS. I hope that my comments will help authors to improve their manuscript.

Detailed remarks concerning the manuscript:

1. The title of the manuscript is inappropriate and did not reflect to the studies presented in the text of the manuscript

2. Key words: It is not recommended to use as key words the words or phrases used in the title of the manuscript. Please do needed changes

3. The Latin names of the species should be italicized. Please go through the whole text of the manuscript and do needed changes.

4. Lines 44-45. “A large variation in Zn accumulation have been previously reported among barley genotypes [e.g., 12,13,14].” Does the abbreviation e.g. before the reference numbers is needed.

5. The treatments should be clearly given in the metodology as well as the number of the repetition in the each treatment.

6. All tables and figures should be clear for the leader without referring to the text of the manuscript. Please do changes whehe needed.

7. Please check whether the tables and figures should be Cite in the text of the manuscript as „Figure”, „Table” or  „Fig.”, „Tab.”

8. „This indicates that cv Sahara and Yerong 81 are probably more tolerant to Zn deficiency than cv Franklin, which showed greater change in plant Zn accumulation due to Zn supply” it should be dot after cv. abbreviation. Do needed chan ges in the whole text of manuscript.

9. Bibliography should be prepared strictly to the guidelines for authors. There are many editorial mistakes in it. There is impossible to mention all of them. There are some examples:

a). Once the abbreviated titles, but the other time the full journal titles are presented. See: ”Tiong, J.; McDonald, G.; Genc, Y.; Shirley, N.; Langridge, P.; Huang, C.Y. Increased expression of six ZIP family genes by zinc 395 (Zn) deficiency is associated with enhanced uptake and root-to-shoot translocation of Zn in barley (Hordeum vulgare). New Phytol. 2015, 207, 1097-1109, doi:10.1111/nph.13413.” and ”Chiou, T.-J.; Aung, K.; Lin, S.-I.; Wu, C.-C.; Chiang, S.-F.; Su, C.-l. Regulation of phosphate homeostasis by microRNA in Arabidopsis. The Plant Cell 2006, 18, 412-421, doi:10.1105/tpc.105.038943”

b). Once each Word in the manuscript title is followed by dot but the oter time not. See: and ”Hussain, S.; Khan, A.M.; Rengel, Z. Zinc-biofortified wheat accumulates more cadmium in grains than standard wheat when grown on cadmium-contaminated soil regardless of soil and foliar zinc application. Sci Total Environ 2019, 654, 402-408, 347 doi:10.1016/j.scitotenv.2018.11.097”. and ”Xu, J.; Qin, X.; Ni, Z.; Chen, F.; Fu, X.; Yu, F. Identification of zinc efficiency-associated loci (ZEALS) and candidate genes for Zn 366 deficiency tolerance of two recombination inbred line populations in maize. Int. J. Mol. Sci. 2022, 23, 4852, doi:10.3390/ijms23094852”

c).  The Journal title Agronomy should not be abbreviated. See: ”Sakellariou, M.; Mylona, P.V. New uses for traditional crops: the case of barley biofortification. Agron. 2020, 10, doi:10.3390/agronomy10121964.” Please give full bibliografic data for this reference.

Please go through the whole references and do needed changes.

10. The proposal for the future studiem should be given.

11. The practical application of the study results should be given.

Author Response

Dear Reviewer,

Thanks for your constructive suggestions. We have now revised the MS according to your suggestions. All revised parts are in red font. Below are responses to your comments.

Reviewer #2:

The title of the manuscript is inappropriate and did not reflect to the studies presented in the text of the manuscript

Reply: The title of the manuscript has been changed to “Mapping QTLs for mineral accumulation and shoot dry biomass in barley under different levels of zinc supply”

Key words: It is not recommended to use as key words the words or phrases used in the title of the manuscript. Please do needed changes

Reply: The key words have been revised according to the reviewer’s suggestion

The Latin names of the species should be italicized. Please go through the whole text of the manuscript and do needed changes.

Reply: It has been rechecked throughout the manuscript and corrected accordingly.

Lines 44-45. “A large variation in Zn accumulation have been previously reported among barley genotypes [e.g., 12,13,14].” Does the abbreviation e.g. before the reference numbers is needed.

Reply: This abbreviation has been deleted now in the revised version.

The treatments should be clearly given in the methodology as well as the number of the repetition in the each treatment.

Reply: There were two treatments used in this experiment: adequate zinc and low zinc levels. This information has been provided in the text (lines 289-292). The details about the number of replicates have been added to the text in the revised version (line 295-297).

All tables and figures should be clear for the reader without referring to the text of the manuscript. Please do changes where needed.

Reply: More details are added to Figure captions.

Please check whether the tables and figures should be Cite in the text of the manuscript as „Figure”, „Table” or  „Fig.”, „Tab.”

Reply: This has been corrected and now the tables and figures are cited appropriately in the text of the manuscript as “Figure, Table”.

This indicates that cv Sahara and Yerong are probably more tolerant to Zn deficiency than cv Franklin, which showed greater change in plant Zn accumulation due to Zn supply” it should be dot after cv. abbreviation. Do needed changes in the whole text of manuscript.

Reply: We corrected the cv. abbreviation accordingly in the entire text of the revised version

Bibliography should be prepared strictly to the guidelines for authors. There are many editorial mistakes in it. There is impossible to mention all of them.

a). Once the abbreviated titles, but the other time the full journal titles are presented. See: ”Tiong, J.; McDonald, G.; Genc, Y.; Shirley, N.; Langridge, P.; Huang, C.Y. Increased expression of six ZIP family genes by zinc 395 (Zn) deficiency is associated with enhanced uptake and root-to-shoot translocation of Zn in barley (Hordeum vulgare). New Phytol2015207, 1097-1109, doi:10.1111/nph.13413.” and ”Chiou, T.-J.; Aung, K.; Lin, S.-I.; Wu, C.-C.; Chiang, S.-F.; Su, C.-l. Regulation of phosphate homeostasis by microRNA in ArabidopsisThe Plant Cell 200618, 412-421, doi:10.1105/tpc.105.038943”

b). Once each Word in the manuscript title is followed by dot but the oter time not. See: and ”Hussain, S.; Khan, A.M.; Rengel, Z. Zinc-biofortified wheat accumulates more cadmium in grains than standard wheat when grown on cadmium-contaminated soil regardless of soil and foliar zinc application. Sci Total Environ 2019654, 402-408, 347 doi:10.1016/j.scitotenv.2018.11.097”. and ”Xu, J.; Qin, X.; Ni, Z.; Chen, F.; Fu, X.; Yu, F. Identification of zinc efficiency-associated loci (ZEALS) and candidate genes for Zn 366 deficiency tolerance of two recombination inbred line populations in maize. Int. J. Mol. Sci. 202223, 4852, doi:10.3390/ijms23094852”

c).  The Journal title Agronomy should not be abbreviated. See: ”Sakellariou, M.; Mylona, P.V. New uses for traditional crops: the case of barley biofortification. Agron. 202010, doi:10.3390/agronomy10121964.” Please give full bibliografic data for this reference.

Reply: Accepted. All the bibliographic mistakes have been corrected.

The proposal for the future studies should be given.

Explained. We did mention this in the text (line 218-219; 249-250).

Round 2

Reviewer 1 Report

Before recommending acceptance of this manuscript, I would ask the authors to consider the minor points below.

P3L87: "Yeong" → Yerong (?)

Supplemental information:

A more accessible format file (word, excel, pdf, etc.) is preferred.

Author Response

Thanks for your careful checking. The word has been corrected and the supplementary files have been replaced by excel format.